

# Identification of high protein kinase CK2α in HPV(+) oropharyngeal squamous cell carcinoma and correlation with clinical outcomes

Janeen H. Trembley[1,2,3,*], Bin Li[4,5,9,*], Betsy T. Kren[1,3], Justin Peltola[2,6], Juan Manivel[2,6], Devi Meyyappan[7,10], Amy Gravely[1], Mark Klein[3,7,8], Khalil Ahmed[1,2,3,5] and Emiro Caicedo-Granados[3,4,5]

[1] Research Service, Minneapolis VA Health Care System, Minneapolis, MN, United States of America

[2] Department of Laboratory Medicine and Pathology, University of Minnesota - Twin Cities Campus, Minneapolis, MN, United States of America

[3] Masonic Cancer Center, University of Minnesota - Twin Cities Campus, Minneapolis, MN, United States of America

[4] Otolaryngology Section, Minneapolis VA Health Care System, Minneapolis, MN, United States of America

[5] Department of Otolaryngology, University of Minnesota - Twin Cities Campus, Minneapolis, MN, United States of America

[6] Laboratory Medicine and Pathology Service, Minneapolis VA Health Care System, Minneapolis, MN, United States of America

[7] Hematology and Oncology Section, Minneapolis VA Health Care System, Minneapolis, MN, United States of America

[8] Department of Medicine, University of Minnesota - Twin Cities Campus, Minneapolis, MN, United States of America

[9] Current affiliation: Kaiser Permanente Roseville Medical Center, Department of Head and Neck Surgery, Roseville, CA, United States of America

[10] Current affiliation: University of Texas Medical Branch, University Blvd, Galveston, TX, United States of America

[*] These authors contributed equally to this work.

Corresponding author
Janeen H. Trembley,
trem0005@umn.edu

## ABSTRACT

**Background**. Oropharyngeal squamous cell carcinoma (OPSCC) incidence is rising worldwide, especially human papillomavirus (HPV)-associated disease. Historically, high levels of protein kinase CK2 were linked with poor outcomes in head and neck squamous cell carcinoma (HNSCC), without consideration of HPV status. This retrospective study examined tumor CK2α protein expression levels and related clinical outcomes in a cohort of Veteran OPSCC patient tumors which were determined to be predominantly HPV(+).

**Methods**. Patients at the Minneapolis VA Health Care System with newly diagnosed primary OPSCC from January 2005 to December 2015 were identified. A total of 119 OPSCC patient tumors were stained for CK2α, p16 and Ki-67 proteins and E6/E7 RNA. CK2α protein levels in tumors and correlations with HPV status and Ki-67 index were assessed. Overall survival (OS) analysis was performed stratified by CK2α protein score and separately by HPV status, followed by Cox regression controlling for smoking status. To strengthen the limited HPV(−) data, survival analysis for HPV(−) HNSCC patients in the publicly available The Cancer Genome Atlas (TCGA) PanCancer RNA-seq dataset was determined for *CSNK2A1*.

**Results**. The patients in the study population were all male and had a predominant history of tobacco and alcohol use. This cohort comprised 84 HPV(+) and 35 HPV(−) tumors. CK2α levels were higher in HPV(+) tumors compared to HPV(−) tumors. Higher CK2α scores positively correlated with higher Ki-67 index. OS improved with increasing CK2α score and separately OS was significantly better for those with HPV(+) as opposed to HPV(−) OPSCC. Both remained significant after controlling for smoking status. High *CSNK2A1* mRNA levels from TCGA data associated with worse patient survival in HPV(−) HNSCC.

**Conclusions**. High CK2α protein levels are detected in HPV(+) OPSCC tumors and demonstrate an unexpected association with improved survival in a strongly HPV(+) OPSCC cohort. Worse survival outcomes for high *CSNK2A1* mRNA levels in HPV(−) HNSCC are consistent with historical data. Given these surprising findings and the rising incidence of HPV(+) OPSCC, further study is needed to understand the biological roles of CK2 in HPV(+) and HPV(−) HNSCC and the potential utility for therapeutic targeting of CK2 in these two disease states.

# INTRODUCTION

The majority of cancers that arise from the mucosal epithelium in the oral cavity, pharynx and larynx are grouped together as HNSCC. A critical categorization that has arisen in HNSCC is the presence or absence of HPV infection, and the bulk of HPV(+) HNSCC derives from the oropharynx (oropharyngeal squamous cell carcinoma, OPSCC) (*Fakhry et al., 2014*; *Johnson et al., 2020*; *Klein & Grandis, 2010*; *Pfister & Fury, 2014*). In recent years, HPV-related OPSCC has been recognized as a unique entity that behaves differently from other HNSCC types connected with traditional carcinogens such as tobacco and alcohol. Notably, HPV(+) OPSCC patients have significantly better overall survival and disease-free survival compared to their HPV(−) counterparts (*Ang et al., 2010*; *Chaturvedi et al., 2011*). HPV status was recently designated as a relevant biomarker, and detection of the cellular protein p16$^{INK4A}$ (*CDKN2A*) serves as a surrogate marker for HPV positivity (*Beltz et al., 2019*; *Johnson et al., 2020*; *Rosenthal et al., 2016*). The malignant transformation of HPV infected cells involves inactivation of tumor suppressor proteins p53 and Rb by viral oncoproteins E6 and E7, respectively (*Gillison et al., 2000*; *Rampias et al., 2009*). The HPV-encoded E6 and E7 RNAs can be detected in tumor tissue to indicate HPV infection.

Protein kinase CK2 is a ubiquitous and highly conserved protein serine/threonine kinase that promotes cell growth and cell proliferation and also suppresses apoptosis. CK2 is a heterotetrameric enzyme consisting of two catalytic α and α′ subunits linked *via* two regulatory β subunits, and these proteins localize to both the cytoplasmic and nuclear compartments (*Faust et al., 1999*). Numerous CK2 substrates have been identified indicating its involvement in a large number of cellular activities (*Borgo et al., 2021*). Further, multiple important cellular signaling pathways are affected by CK2 activity,

including, but not limited to, NFκB, AKT, TP53, Wnt/β-catenin and PTEN (*Dominguez, Sonenshein & Seldin, 2009*; *Ruzzene & Pinna, 2010*; *Trembley et al., 2010*).

CK2 has been found to be elevated relative to normal tissue in nearly all cancer types, including head and neck squamous cell carcinoma (HNSCC) in general as well as OPSCC (*Chua, Lee & Dominguez, 2017*; *Trembley et al., 2009*). Previous studies established that elevated CK2 activity or levels in head and neck cancer are associated with Ki-67-positive tumor cells, aggressive tumor behavior and poor clinical outcome (*Chua, Lee & Dominguez, 2017*; *Faust et al., 1996*; *Faust et al., 1999*; *Gapany et al., 1995*); however, these studies did not incorporate tumor HPV status. CK2 phosphorylates and/or regulates HPV proteins such as E1 and E7 and cellular proteins such as Brd4 to exert influence over HPV DNA replication and cellular proliferation (*Basukala et al., 2019*; *Firzlaff et al., 1989*; *Iftner et al., 2017*; *Piirsoo et al., 2019*; *Zine El Abidine et al., 2017*). Therefore, we postulate that CK2 is likely to play a role in the pathogenesis of HPV-related HNSCC and may potentially influence tumor behavior.

To our knowledge, this is the first report on the expression of CK2α in HPV(+) OPSCC tumors. In the present retrospective study we examined tumor CK2α and Ki-67 protein levels and HPV status. First, we describe the relationships between CK2α protein levels in tumors and patient characteristics, HPV status, and Ki-67 index. We present overall survival (OS) analyses stratified by CK2α protein level and separately by HPV status. We examine other univariate relationships with survival to identify covariates to control for in our two main analyses of interest. Finally, we evaluated survival outcomes in HPV(−) HNSCC using The Cancer Genome Atlas (TCGA) RNA-seq data.

## MATERIALS AND METHODS

### Patients

The study was conducted according to the guidelines of the Declaration of Helsinki, and approved by the Institutional Review Board of Minneapolis VA Health Care System (4632-A, approved June 6, 2016). Patient consent was waived by the Institutional Review Board due to the retrospective nature of the study. Data analysis was completed under an approved Research and Development Committee protocol (VAM-20-00609, approved July 17, 2020).

For this retrospective cohort study, a database search was performed in the tumor registry for the Minneapolis Veterans Affairs Healthcare System. All patients with newly diagnosed primary OPSCC from January 2005 to December 2015 were identified. Inclusion criteria were (1) biopsy-proven primary OPSCC, and (2) adequate tissue in archived tissue bank to perform immunohistochemical (IHC) staining. Patients who had other simultaneous active malignancy at the time of diagnosis or lacked follow up were excluded. Patient demographics including age at diagnosis, gender, smoking history, and alcohol consumption history were recorded. Tumor characteristics including primary subsite, tumor stage, and treatment modalities were reviewed from patient charts. Clinical outcomes including disease progression and death were also recorded. Patients were censored at the end of this study.

### p16, Ki-67 and CK2 IHC stain

Formalin-fixed paraffin-embedded (FFPE) tissue blocks were retrieved from the pathology tissue archive. IHC stain was performed with Bond Refine Polymer Detection Kit on the Bond III Automated Stainer (Leica Biosystems, Buffalo Grove, IL, USA), with the following primary antibodies: mouse monoclonal anti-p16 antibody E6H4 (Ventana Medical Systems, Oro Valley, AZ, USA), rabbit monoclonal anti-Ki-67 antibody SP6 (Cell Marque, Sigma-Aldrich, Rocklin, CA, USA), and rabbit monoclonal anti-CK2 alpha antibody ab76040 (Abcam, Cambridge, United Kingdom). Positive p16 expression was defined as greater than 70% of tumor cells with strong nuclear and cytoplasmic staining. Reactivity for Ki-67 was evaluated as percentage of viable tumor nuclei positive for this marker.

CK2α staining was verified using myometrium for the "negative" control which showed a very low level staining and using testicle as the positive control as it has a high level of CK2. For further negative control, the CK2α primary antibody was substituted with appropriate isotype control antibody ab172730 (Abcam). Specificity of CK2α staining was tested by comparing CK2α antibody to isotype control in sequential sections from five OPSCC tumors with CK2α score of 3 and five OPSCC tumors with CK2α score of 1. CK2α scoring was evaluated by semi-quantitative assessment of the relative antigen density in combined cytoplasm and nucleus of viable tumor cells (*Zhou et al., 2014*). Reactivity was evaluated as score 3 (intense), score 2 (moderate), score 1 (weak), and score 0 (no staining). All scoring and expression levels were evaluated and agreed upon by two staff pathologists (J.P. and J.C.M.).

### E6 and E7 RNA *in situ* hybridization stain

E6 and E7 mRNA of HPV16/18 were detected in tumor sections using the Bond RNAscope kit and following the Leica RNAscope DAB *in situ* hybridization protocol (*RS_DAB, Leica Biosystems). Processing was performed using the Automated Stainer instrument.

### Definition of HPV status

To obtain unambiguous information on the HPV status in these tumors, they were stained for cellular p16[INK4a], commonly used as a surrogate marker for HPV positivity, and for the HPV RNAs E6 and E7. P16 was positive in 70.3% of patients and E6/E7 RNAs were detected in 70.6% of patients. Two patient tumors were p16-positive and E6/E7 RNA-negative; conversely, two tumors were E6/E7 RNA-positive and p16-negative. As expected, the relationship between p16 and E6/E7 detection in tumors was significant ($P < 0.0001$). In the remainder of this study, positive E6/E7 detection in tumors was used as our operational definition of HPV(+) *vs.* HPV(−) status, including those patients with discordant p16 results.

### Human HNSCC TCGA data analysis

Analysis of survival associated with *CSNK2A1* mRNA levels in HNSCC HPV(−) patient samples was performed *via* cBioPortal using the PanCancer Atlas data set from The Cancer Genome Atlas ($n = 415$ HPV(−) samples for OS and PFS; $n = 394$ for DSS; 23 September 2021). For mRNA expression data, TCGA typically computes the relative expression of

an individual gene in a tumor sample to the gene's expression distribution in a reference population of samples. In this analysis, the reference set was all samples that are diploid for *CSNK2A1*. The returned value indicates the number of standard deviations away from the mean of expression in the reference population ($Z$-score) $Z$-score cut-off was set at $>+1.5$ higher expression of *CSNK2A1* compared to unaltered or lower levels of *CSNK2A1*.

## Statistical analysis

In Table 1, we utilize counts and proportions for categorical variables to describe the sample. We present descriptive data overall and examine the relationships between CK2 score category and the variables HPV status or Ki-67. Analyses for Table 2 included Pearson's chi-square test for categorical variables or Fisher's exact test where any cell size was less than 5. The chi-square test for independence was used to test the relationship between positive detection of E6/E7 RNA and of p16. We defined statistical significance by *P* value < 0.05. The analyses were performed in SAS 9.4® software.

## Survival analysis

Utilizing the Veterans cohort dataset, univariate Cox regression was performed for clinical variables CK2α and HPV status (predictors of interest) and for co-variates Ki-67, treatment, age, alcohol, smoking, T and N classifications, and clinical stage (7th edition of the AJCC cancer staging manual) to explore if any of these should be controlled for in our main two survival analyses of interest (Table 3). As our main analysis of interest, OS was stratified by HPV and separately by CK2α. Survival curves for OS according to CK2α score and separately for HPV status were fitted using the Kaplan–Meier method. Log rank *P*-values are presented. Since smoking status (current *vs.* never or former) was significant by univariate analysis, we additionally controlled for this in multivariate Cox regressions. OS time was defined as from treatment initiation to death. PFS time was defined as from treatment initiation to disease progression or death. These analyses were performed in SAS 9.4® software. Analyses using the TCGA cohort dataset were performed using cBioPortal. A *P*-value of 0.05 was utilized to denote statistical significance.

# RESULTS

## Patient and tumor characteristics overall

A total of 147 patients at the Minneapolis VA hospital between 2005 and 2015 met the inclusion criteria, and sufficient tumor tissue was available for analysis in 119 patients. The mean age of diagnosis for all patients, as well as the HPV(+) and HPV(−) subgroups, was 63 years. The diagnosis age range for HPV(+) was 44 to 88 years, and for HPV(−) was 53 to 82 years. As shown in Table 1, the majority of the patients had a history of smoking (94.1%) and alcohol use (84.0%). The most common subsite of OPSCC was tonsil (58.6%), followed by base of tongue (35.3%). The majority of the patients presented with stage IV disease (70.6%) according to the 7th edition of the AJCC cancer staging manual. All tumors were conventional type squamous cell carcinoma histology.

All patients received treatment compliant with NCCN guidelines and with curative intent. Our data show that 70% of patients completed treatment for all modalities. Breaking

**Table 1  OPSCC patient characteristics.**

|  | n | (Percentage) |
|---|---|---|
| Gender |  |  |
|   Male | 119 | (100) |
|   Age | 63 ± 7.2 |  |
| Smoking status |  |  |
|   Current | 58 | (48.7) |
|   Former | 54 | (45.4) |
|   Never | 7 | (5.9) |
| Alcohol Use |  |  |
|   Current | 58 | (48.7) |
|   Former | 42 | (35.3) |
|   Non-drinker | 19 | (16.0) |
| Tumor Site |  |  |
|   Tonsil | 68 | (58.6) |
|   Base of Tongue | 41 | (35.3) |
|   Soft Palate | 6 | (5.2) |
|   Pharyngeal Wall | 1 | (0.9) |
| T Classification |  |  |
|   T1 | 46 | (38.7) |
|   T2 | 39 | (32.8) |
|   T3 | 19 | (16.0) |
|   T4 | 15 | (12.6) |
| N Classification |  |  |
|   N0 | 24 | (20.2) |
|   N1 | 14 | (11.8) |
|   N2 | 72 | (60.5) |
|   N3 | 9 | (7.5) |
| TNM Stage (AJCC) |  |  |
|   I | 10 | (8.4) |
|   II | 9 | (7.6) |
|   III | 16 | (13.4) |
|   IVA | 73 | (61.3) |
|   IVB | 11 | (9.2) |
| Initial Treatment |  |  |
|   Surgery | 12 | (10.1) |
|   Radiation | 12 | (10.1) |
|   CCRT | 69 | (58.0) |
|   Surgery/Radiation | 6 | (5.0) |
|   Surgery/CCRT | 20 | (16.8) |

**Notes.**
CCRT,  concurrent chemoradiotherapy.

**Table 2  OPSCC tumor data according to CK2α score.**

|  | | Total | CK2α Score 1 (% of total) | | CK2α Score 2 (% of total) | | CK2α Score 3 (% of total) | | P-value |
|---|---|---|---|---|---|---|---|---|---|
| **n** | | **119** | 28 | (23.5) | 60 | (50.4) | 31 | (26.1) | |
| **HPV status** | | | | | | | | | |
| Positive | 84 | (70.6) | 6 | (21.4) | 50 | (83.3) | 28 | (90.3) | <0.0001[*] |
| Negative | 35 | (29.4) | 22 | (78.6) | 10 | (16.7) | 3 | (9.7) | |
| **Ki-67** | | | | | | | | | |
| Low (<10%) | 6 | (5.0) | 5 | (17.9) | 1 | (1.7) | 0 | (0) | 0.0003[a] |
| Borderline (10–20%) | 27 | (22.7) | 11 | (39.3) | 11 | (18.3) | 5 | (16.1) | |
| High (>20%) | 86 | (72.3) | 12 | (42.9) | 48 | (80.0) | 26 | (83.4) | |

Notes.
[*]Fisher's exact test.
[a]Pearson's chi-square.

**Table 3  Cox univariate regression analysis for overall survival.**

| Variable | | HR (95% CI) | P Value |
|---|---|---|---|
| CK2α | High (score 2 & 3) *vs.* low (score 1) | 0.398 (0.214, 0.738) | 0.0035 |
| HPV | Negative *vs.* Positive | 3.128 (1.692, 5.783) | 0.0003 |
| Ki-67 | Low (≤20%) *vs.* high (>20%) | 1.587 (0.834, 3.020) | 0.1596 |
| Age | Per year increase | 1.011 (0.970, 1.053) | 0.6095 |
| Alcohol | current/former *vs.* non-drinker | 1.655 (0.590, 4.641) | 0.3380 |
| Smoking | Current *vs.* former/never | 2.359 (1.232, 4.517) | 0.0096 |
| T Classification | T3-4 *vs.* T1-2 | 1.304 (0.685, 2.481) | 0.4192 |
| N Classification | N1, N2, & N3 *vs.* N0 | 1.259 (0.580, 2.730) | 0.5601 |
| TNM Stage (AJCC) | Per increase in stage | 1.152 (0.852, 1.558) | 0.3582 |
| Treatment | CCRT vs. Surgery | 1.300 (0.451, 3.747) | 0.6269 |
| | Surgery/CCRT *vs.* Surgery | 0.584 (0.162, 2.109) | 0.4116 |
| | Surgery/Radiation *vs.* Surgery | 0.438 (0.049, 3.950) | 0.4619 |
| | Radiation *vs.* Surgery | 1.407 (0.376, 5.270) | 0.6122 |

Notes.
CCRT, concurrent chemoradiation.

down the different treatment groups shown in Table 1, the percentages of patients that completed the full treatment regimen are as follows: Surgery 100%, Radiation 75%, CCRT 68%, Surgery/Radiation 83%, and Surgery/CCRT 50%.

## CK2α staining and association with patient characteristics

Tumors were stained for CK2α protein, and CK2α scoring was evaluated by semi-quantitative assessment of the relative antigen density in combined cytoplasm and nucleus of viable tumor cells. Reactivity was evaluated as score 3 (intense), score 2 (moderate), score 1 (weak), and score 0 (no staining). As expected, due to ubiquitous expression of CK2α in most cancer cells, no scores of 0 were observed for CK2α. Representative images of CK2α score 1, 2 and 3 with corresponding H&E and Ki-67 stains are depicted in Fig. 1. The percentage of patients with CK2α tumor score of 1, 2 and 3 were 23.5%, 50.4% and 26.1%, respectively. Age at diagnosis, alcohol use, T or N classification, or treatment modality did

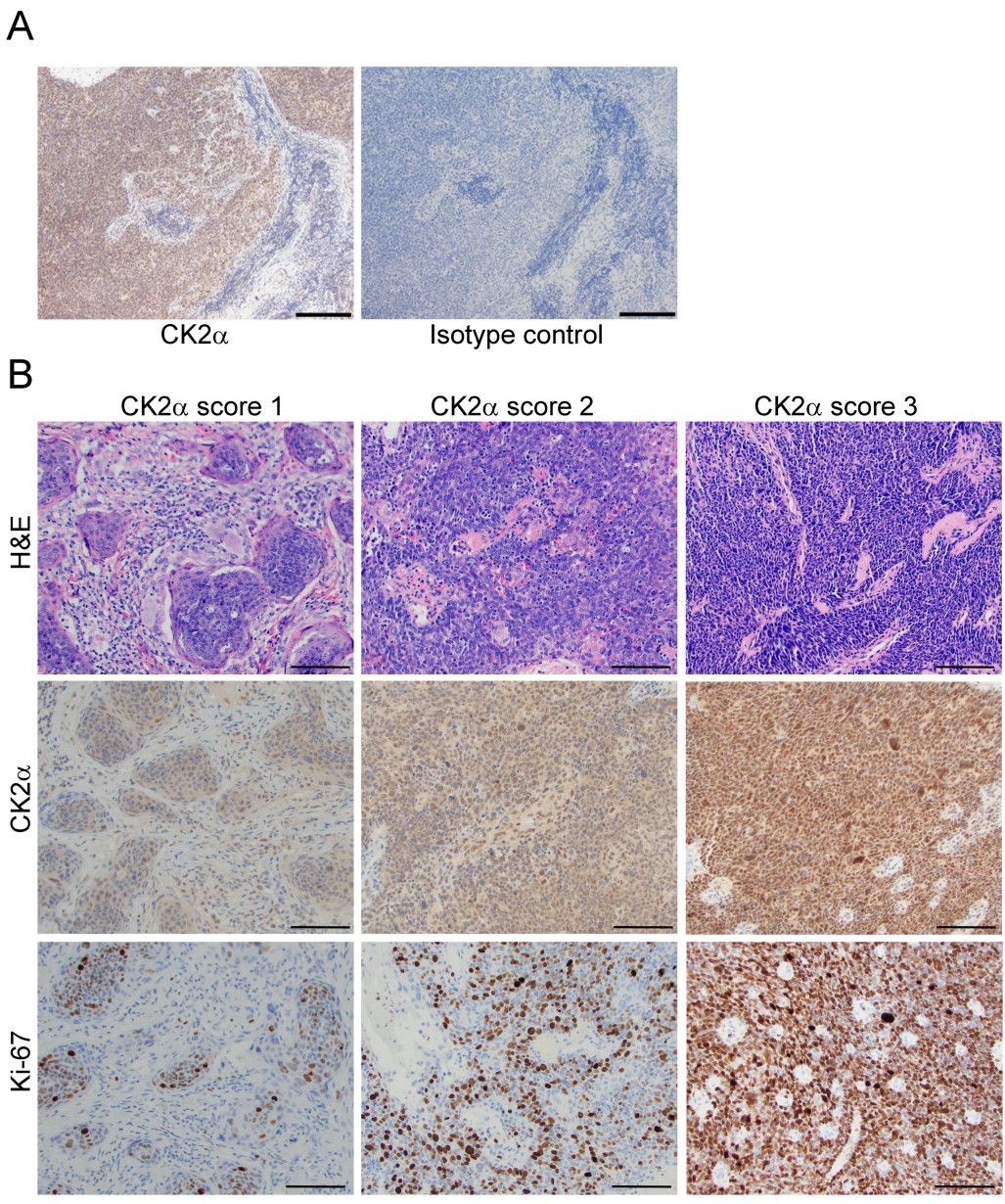

**Figure 1** **Expression of CK2α in OPSCC tumors.** (A) Representative field following CK2α *versus* isotype control antibody IHC stain in OPSCC tumor tissue is shown. CK2α was detected by DAB (brown). Blue depicts nuclei. Identity of staining antibody is indicated below each panel. Scale bar, 100 μm. (B) Representative fields of CK2α IHC stain in OPSCC tumor tissues are shown with corresponding H&E and Ki-67 stain in the same tumors. CK2α and Ki-67 were detected by DAB (brown). Blue depicts nuclei. Identity of stain is indicated at the left and CK2α score is indicated above the panels. H&E, hematoxylin and eosin. Scale bar, 100 μm.

not differ significantly between CK2α score categories. The majority of stage I tumors had CK2α score of 1.

### Evaluation of CK2α protein levels in tumors and correlations with HPV status and Ki-67 index

HPV(+) tumors, as defined by positive detection of E6/E7 RNA, represented 21.4% of CK2α score 1 tumors, 83.3% of CK2α score 2 tumors, and 90.3% of CK2α score 3 tumors. We examined whether CK2α expression levels and HPV positivity were associated. As shown in Table 2, CK2α score strongly corresponded with HPV(+) OPSCC ($P < 0.0001$).

Given the different mutational and HPV statuses of these malignancies, we were also interested in the proliferative state of these tumors and thus performed Ki-67 staining and analysis. We found that the tumors in this entire cohort were highly proliferative, with 72.3% of tumor cells more than 20% positive for Ki-67. In analysis of CK2α scores with the Ki-67 index in tumors, there was a positive relation of higher CK2α score with higher percentage of Ki-67 positivity ($P < 0.0003$; Table 2).

### Survival analysis

PFS rates and OS rates among the CK2α scores were very similar, and we focused on analysis of OS. Univariate Cox regression analysis for OS was performed for the two independent variables of interest CK2α and HPV and the potential co-variates Ki-67, treatment, age, alcohol, smoking, TNM stage, and clinical classification (7th edition of the AJCC staging classification) to investigate their association with survival time of patients and potential to be entered into multivariate Cox regression (Table 3). The relationships CK2α (high score of 2 or 3 *vs.* low score of 1) and HPV status (independent variables of interest) and co-variate smoking (current *vs.* former or never) were all statistically significant. Kaplan–Meier analysis comparing the three CK2α scores in all patients showed that OS was significantly different with improved OS as CK2α score increased (Fig. 2A). Kaplan–Meier analysis by HPV status (positive *vs.* negative) in all patients demonstrated significantly better OS for HPV(+) OPSCC (Fig. 2B).

Because the co-variate smoking was significant in univariate analysis, we controlled for smoking in two separate multivariate Cox regressions for the predictors CK2α or HPV. CK2α level (high score of 2 or 3 *vs.* low score of 1) or HPV status (negative *vs.* positive) remained statistically significant after controlling for smoking (CK2α HR 0.461 [0.246, 0.867], $P = 0.0162$; HPV HR 2.567 [1.329, 4.959], $P = 0.0050$).

### Survival analysis in TCGA HPV(−) HNSCC

Our group of HPV(−) OPSCC patients is limited, but previous studies suggest that high CK2 levels (α, α′, β) in the context of HPV(−) disease are a risk for worse outcomes (*Chua et al., 2017*; *Faust et al., 1996*; *Gapany et al., 1995*). Because the number of patients with HPV(−) disease was low in our cohort, we analyzed the publicly available TCGA HNSCC PanCancer RNA-seq data using cBioportal (*Cerami et al., 2012*; *Gao et al., 2013*). Disease-specific survival analysis was performed in the HPV(−) cohort comparing high *CSNK2A1* (CK2α) mRNA levels (*Z*-score cut-off $> 1.5$, $n = 117$) to unaltered or low *CSNK2A1* mRNA ($n = 277$). In HPV(−) HNSCC, high *CSNK2A1* mRNA levels associated

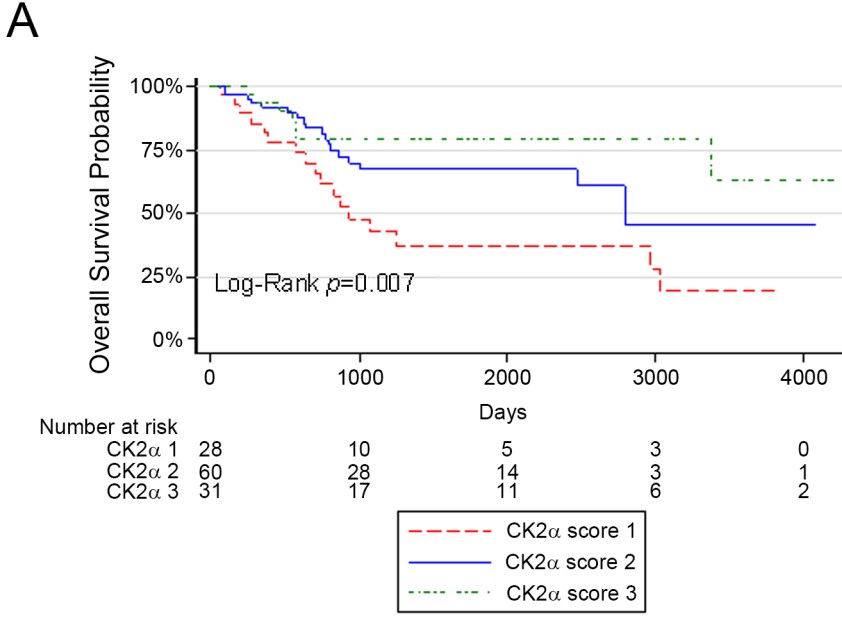

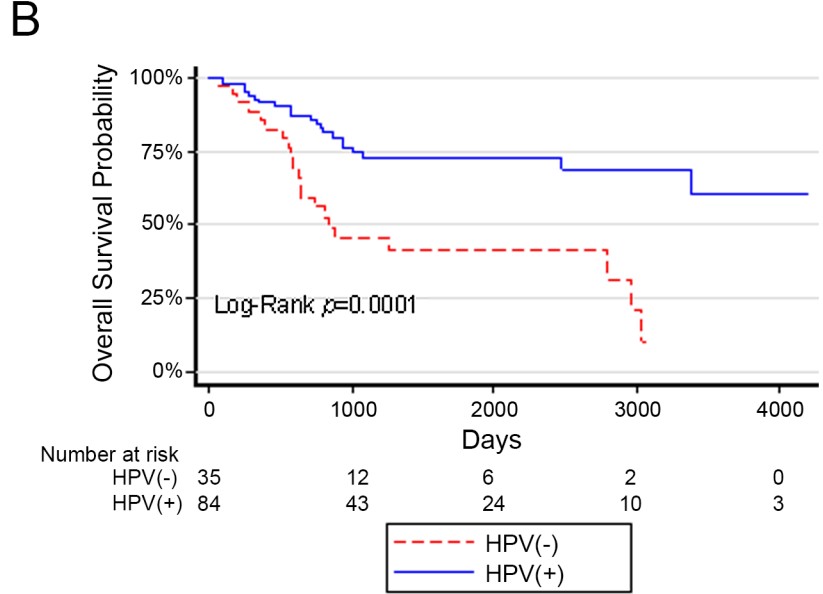

**Figure 2  Survival curves based on CK2 scores or HPV status in Minneapolis VA cohort.** Kaplan–Meier survival curves are shown. (A) OS according to CK2α score. (B) OS according to HPV status. The legend identifying each survival curve line is indicated below each chart.

with significantly worse survival (logrank $P = 0.0279$; $q = 0.0478$; Fig. 3). PFS and OS were also significantly worse for high *CSNK2A1* (logrank $P = 0.0314$ and $P = 0.0359$, respectively; $q = 0.0478$ for both PFS and OS). In the HPV(+) cohort, the subset of patients with high *CSNK2A1* mRNA tumors consisted of only 7 tumors relative to 56 tumors with

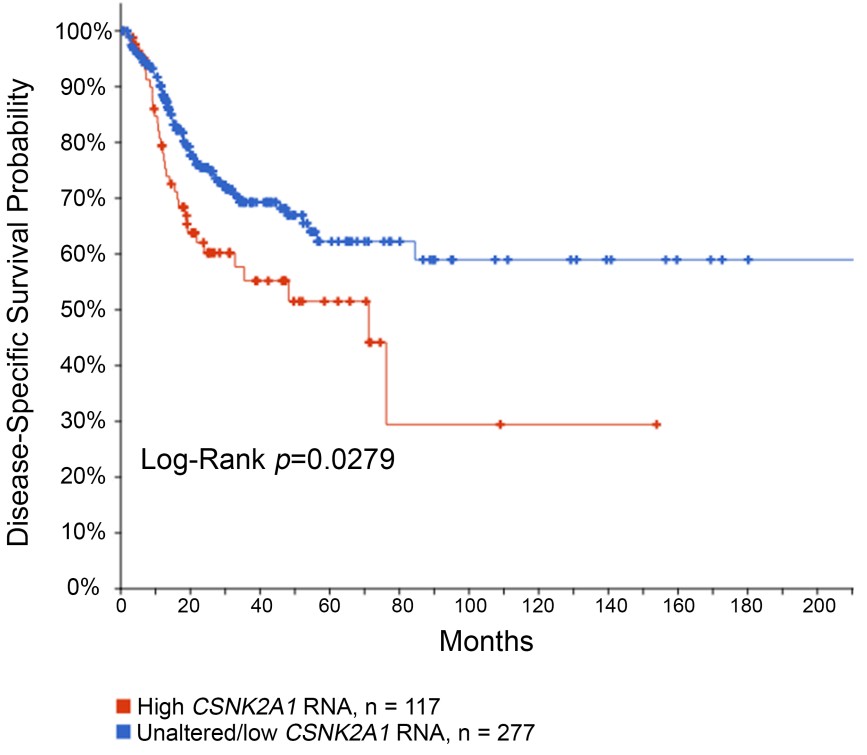

**Figure 3** **Disease-specific survival based on *CSNK2A1* mRNA levels in HPV(−) HNSCC.** Kaplan–Meier survival curve is shown for CSNK2A1 (CK2α) mRNA $z$-scores in HPV(−) HNSCC patients from TCGA RNA-seq data. The legend is indicated below the chart.

unaltered *CSNK2A1*; there was insufficient survival data in the *CSNK2A1* high group for analysis.

## DISCUSSION

In the present work, we have discovered an unexpected difference in the CK2-related survival of HPV(+) compared with HPV(−) HNSCC. This novel aspect is discussed as follows.

Elevated CK2 RNA, protein or enzymatic activity are historically associated with poor clinical outcome in various types of malignancies, including HNSCC (*Chua, Lee & Dominguez, 2017*; *Faust et al., 1996*; *Gapany et al., 1995*; *Kim et al., 2007*; *Laramas et al., 2007*; *Li et al., 2017*; *Lin et al., 2010*; *Lin et al., 2011*; *O-Charoenrat et al., 2004*). In these earlier studies, HPV status was not evaluated in the context of CK2 activity. As the current increase in OPSCC incidence is considerably linked to HPV presence, we undertook this examination of CK2 level and patient outcomes in relation to HPV status. Using RNA-seq data from The Cancer Genome Atlas (TCGA), we found that high *CSNK2A1* (CK2α) mRNA levels associate with worse disease-specific survival, PFS, and OS in HPV(−) HNSCC. Protein and mRNA abundances roughly correlate in cells and tissue, and *CSNK2A1* mRNA levels are significantly elevated in OPSCC

(*Chua, Lee & Dominguez, 2017*; *Schwanhäusser et al., 2011*). Thus, these HPV negative-specific RNA-based data support previous findings that high CK2 levels associate with poor prognosis and survival outcomes in non-HPV related cancers.

In contrast to the above discussion, in this study we found that high CK2α protein level was not associated with worse survival in OPSCC. Rather, in a cohort of OPSCC patients with predominantly HPV(+) disease, we found that high CK2α abundance associated with improved survival outcome. It is well documented that HPV(+) OPSCC patients have improved survival, and the data show that high CK2α protein levels are present in the tumors of HPV(+) OPSCC patients. These are interesting indications that CK2 may play specialized roles in HPV(+) HNSCC, and accord with published information on CK2 regulation of papillomavirus lifecycle.

The HPV E6 and E7 oncoproteins promote immortalization and transformation in infected cells through inactivation of p53 and pRb and related pathway proteins, respectively (*Eckhardt et al., 2018*). Previous studies demonstrated key regulation of E7 function by CK2 (*Basukala et al., 2019*; *Firzlaff et al., 1989*; *Nogueira et al., 2017*; *Zine El Abidine et al., 2017*). For example, the phosphorylation of E7 by CK2 is essential to promote Rb-related p130 degradation and cell cycle S-phase entry (*Chien et al., 2000*; *Genovese et al., 2008*). CK2α is required for HPV DNA replication by regulating the stability and nuclear retention of E1, and CK2 has been proposed as a promising target for the development of anti-viral drugs, including SARS-CoV-2 (*Bouhaddou et al., 2020*; *Cruz et al., 2021*; *Piirsoo et al., 2019*). It has been shown that targeting of CK2 activity using an investigational peptide inhibitor is effective in treating cervical cancer, which is almost entirely HPV(+) (*Sarduy et al., 2015*; *Solares et al., 2009*). This CK2 inhibitor, CIGB-300, was identified in a screen of peptides which bind and block phosphorylation of an HPV16 E7 fusion protein (*Perea et al., 2004*).

Our observation of better OS in HPV(+) OPSCC is consistent with previous reports (*Ang et al., 2010*; *Chaturvedi et al., 2011*). At present, how the higher HPV viral load and viral oncoprotein function correlate with improved patient survival in OPSCC remains unclear (*Cohen et al., 2008*). Here we report high CK2α levels associate with improved survival in OPSCC, using a cohort of patients with predominance of HPV(+) disease. In our cohort of patients, a high percentage of HPV(+) patients were observed in the high CK2α score subgroups (score of 2 or 3). These associations raise the question on the potential cross-regulatory nature of HPV and CK2 interactions. It is unknown if active HPV infection and the downstream effectors have an impact on the level or activity of CK2 itself in OPSCC. We posit that, under conditions of HPV infection, CK2 protein levels rise and that viral and/or cellular protein complexes could divert the function of CK2 proteins to promote the HPV lifecycle as opposed to other pro-survival cancer cell roles.

There is no literature in head and neck cancer investigating CK2 in relation to smoking or alcohol. Our results demonstrated that CK2α level and HPV status remained significant to OS after controlling for smoking as a covariate. In this analysis, we controlled for current smokers *vs.* former/never smokers, which is an important distinction for survival in this disease. The validity of this preliminary finding would require testing on a more heterogeneous patient population in terms of smoking history as 94% of our VA population were former or current smokers. Use of alcohol and tobacco are known factors contributing

to worse outcomes in both HPV(+) and HPV(−) HNSCC; however, there are conflicting data. Large-scale clinical trials reported tobacco exposure to be an important prognostic factor, independent of HPV status (*Ang et al., 2010*; *Gillison et al., 2012*). A recent HPV-blind study using data from the entire VHA health care system from 2000 to 2012 reported significantly worse 2-year OS in ever-smoker Veterans compared to never-smoker Veterans (*Zevallos et al., 2016*). Another study observed that HPV(+) status did not confer better survival in smokers (*Hafkamp et al., 2008*). Two reports in 2015 showed better survival in p16(+) Veterans, despite the high tobacco and alcohol use (*Sandulache et al., 2015*; *Shay et al., 2015*). Further understanding of HPV(+) OPSCC biology and outcomes in the Veterans population is needed (*Sandulache et al., 2020*).

For this observational retrospective study, the potential influences of bias or confounders on the results need to be considered. The patients in the study population were Veterans, all from the same hospital, all male, and had a predominant history of tobacco and alcohol use. Lifestyle and systematic differences between HPV(+) and HPV(−) patients and their treatment regimens are not factors incorporated in the analysis of this study, and these factors could be contributors to OS. Therefore, the findings in this study may not be fully applicable to women or the general population. Similarly, patients who received surgery as the only modality of treatment had lower stage of disease (small primary tumor and small neck lymph nodes), therefore comparison of surgery alone to other treatment modalities incorporates bias in the survival analysis. The relatively small sample size in the HPV(+)/CK2α score 1 subgroup and in the entire HPV(−) cohort limited the statistical power of the study. We acknowledge these potential confounders in the interpretation of this data.

## CONCLUSIONS

The increasing prevalence of HPV-associated head and neck cancers underpins the need to understand their biology and related outcomes. To our knowledge, this is the first report evaluating CK2α expression level and its correlation with either HPV status or survival outcomes in OPSCC patients. CK2α protein levels showed direct positive correlation with HPV positivity. Unexpectedly, high CK2α levels were associated with significantly improved survival in all OPSCC patients, where more than 70% of the cases were HPV(+). Overall, our data demonstrate a surprising and intriguing association between CK2α expression in HPV(+) OPSCC and survival; further studies are necessary to determine if this association has prognostic value. Phase 1 and 2 clinical trials have demonstrated that the oral CK2 inhibitor Silmitasertib/CX-4945 is safe for use in cancer patients, slowing disease progression and extending treatment benefit for some patients with advanced solid tumor cancers. In cultured cells, we have shown that CX-4945 is an effective treatment alone and in combination with cisplatin in both HPV(+) and HPV(−) HNSCC (*Trembley et al., 2021*). Given the high levels of CK2α protein in HPV(+) tumors as well as the association of high CK2α mRNA levels with poor outcomes in HPV(−) patients, our results suggest that future investigation into the incorporation of CX-4945 into treatment strategies is warranted. In addition, the function of CK2 in HPV biology as it pertains to HNSCC requires further study given the emergence of HPV(+) HNSCC.

## ACKNOWLEDGEMENTS

K.A. holds the title of Senior Research Career Scientist awarded by the US Department of Veterans Affairs. We thank VA Pathology Service members Wendy Larson and Sue Dachel for technical assistance.

### Funding

This work was supported by Merit Review research funds I01 BX003282 and I01 BX005091 awarded by the United States Department of Veterans Affairs Biomedical Laboratory Research and Development Service (Khalil Ahmed); Lion's award (Bin Li); private donation (Emiro Caicedo-Granados). There was no additional external funding received for this study. The funders had no role in study design, data collection and analysis, decision to publish, or preparation of the manuscript.

### Grant Disclosures

The following grant information was disclosed by the authors:
Merit Review research funds awarded by the United States Department of Veterans Affairs Biomedical Laboratory Research and Development Service: I01 BX003282, I01 BX005091.
Lion's award.
Private donation.

### Competing Interests

The authors declare that they have no competing interests.

### Author Contributions

- Janeen H. Trembley conceived and designed the experiments, analyzed the data, prepared figures and/or tables, authored or reviewed drafts of the paper, and approved the final draft.
- Bin Li conceived and designed the experiments, performed the experiments, analyzed the data, prepared figures and/or tables, authored or reviewed drafts of the paper, and approved the final draft.
- Betsy T. Kren and Khalil Ahmed conceived and designed the experiments, authored or reviewed drafts of the paper, and approved the final draft.
- Justin Peltola performed the experiments, analyzed the data, prepared figures and/or tables, authored or reviewed drafts of the paper, and approved the final draft.
- Juan Manivel performed the experiments, analyzed the data, authored or reviewed drafts of the paper, and approved the final draft.
- Devi Meyyappan performed the experiments, authored or reviewed drafts of the paper, and approved the final draft.
- Amy Gravely analyzed the data, prepared figures and/or tables, authored or reviewed drafts of the paper, and approved the final draft.
- Mark Klein and Emiro Caicedo-Granados conceived and designed the experiments, analyzed the data, authored or reviewed drafts of the paper, and approved the final draft.

## Human Ethics

The following information was supplied relating to ethical approvals (i.e., approving body and any reference numbers):

The study was approved by the Institutional Review Board of Minneapolis VA Health Care System. Data analysis was completed under an approved Research and Development Committee protocol.

## Data Availability

The deidentified clinical and biomarker data, complete IHC images and statistical analyses are available in the Supplementary Files.

## Supplemental Information

Supplemental information for this article can be found online at http://dx.doi.org/10.7717/peerj.12519#supplemental-information.

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
