# Peer review of "Identification of high protein kinase CK2α in HPV(+) oropharyngeal squamous cell carcinoma and correlation with clinical outcomes"

_PeerJ, doi:10.7717/peerj.12519_

## Round 0.1 · original submission · Major Revisions

As you will see, all reviewers were positive but have identified some issues of clarity that require attention. Please address all points ahead of a revision and detail your response in rebuttal.

The reason I have indicated 'major revision' comes from confounding: Could bias or confounding explain the results? As this has been raised by a statistical expert, I would like you to carefully and fully address this point in rebuttal/revision.

Thanks for submitting this interesting study.

Reviewer 1 ·

Basic reporting

Clear writing throughout.
Appropriate references.
Excellent presentation of the data with respect to figures and tables.
Clear hypothesis; clear results; clear relationship between hypothesis and results.

Experimental design

Straightforward retrospective design.
Clear methodology.
Clear IHC and ISH methods.
Clear statistical design and methods.
Well defined research question and hypothesis regarding the interaction between HPV and CK2a.
Original research question and findings within the Aims and Scope of the journal.

Validity of the findings

Novel description of the interaction between HPV status and CK2a as a secondary function of Ki-67. Good data overall, will need validation in subsequent cohorts by an excellent start and contribution to the literature. Robust, statistically sound analysis. Clear and reasonably phrased conclusions.

Additional comments

Trembley et al. discuss the relationship between CK2alpha and oropharyngeal cancer (OPC) outcomes in a Veteran population with a focus on interactions between CK2a and HPV oncogenic drivers in this complex disease. This is a novel study that is well done overall. Good overall cohort size; combined with TGCA dataset. The large number of HPV+ tumors is a good complement to the existing limited datasets. Good correlation of p16 IHC and E6/E7 ISH.
Some clarification is needed with respect to treatment. Presumably all patients received curative intent treatment? How many of the patients completed treatment compliant with NCCN guidelines? This is important since this will impact survival and may skew the correlations with any individual biomarker.
Overall the analysis is clear, well done and compelling. Obviously this will require additional validation in additional HPV+ cohorts but overall this is an important preliminary dataset that will be of value to the field. The images are clear, the tables are clear and well organized. The text is well organized and of appropriate length.

Reviewer 2 ·

Basic reporting

The article is well written and easy to follow. I have a couple of comments which I hope would add additional clarity to the article.

Reviewer Comments:
==================

Page 12 Line 157/158.
==================
(R = .92, p < 0.0001)

From preceding lines in the paper P16 and E7/E6 RNA are either positive or negative. What hypothesis test does the R and p value refer to? Is this a correlation coefficient (and if so which one)?


Page 12 Section 2.5
================
“Z-score cut-off was set at >1.5 higher expression of 165 CSNK2A1 compared to unaltered levels of CSNK2A1.”

Question: What exactly does the Z-score refer to? It a hypothesis test for gene expression? Some additional details would help clarify this.

Experimental design

This was a retrospective cohort study including all newly diagnosed primary OPSCC from January 2005 to December 2015.
Exclusions included patients who had other simultaneous active malignancy at the time of diagnosis or lacked follow up.
A limited number of variables were collected:
1. Demographics: age at diagnosis, gender, smoking history, alcohol consumption history.
2. Tumor related variables: primary subsite, tumor stage, and treatment modalities
3. Outcome related variables.
Reviewer Comments:
==================
Why would a patient lack follow-up – for example did they move to another treatment centre? Can you provide any characterisation of why they weren’t followed-up?

Was there a limited number of variables available on the patients? I’m thinking here of co-morbidities (renal or liver for example), co-medications etc. If these were available why were they not included in the analysis in some form to give a broader idea of the patients?
Can you please state explicitly how censoring was handled – I’m assuming that patients were censored at the end of the study, is this correct?

Page 12 Line 171
==============
What was the structure of the logistic regression equation and could you put in a reference to where it is reported please?


Page 13 Line 184/185
==================
“OS time was defined as from treatment initiation to death. PFS time was defined as from treatment initiation to disease progression.”
Did any patients die before treatment commenced and if so, how where they handled in the analysis? Was there an untreated group?

Validity of the findings

I feel that the paper has been caught between a possible study of the mechanics of the disease / interaction with the HPV, and an observational study of the disease in a particular cohort. And this is the paper’s weakness. I will make some initial comments first and then address where I think the paper has issues that need to be fully addressed.

Some initial comments:
====================
Can you please specify that the age in Table 1 is the median age and add the IQR or range.

Page 14 Line 182
==============
“In analysis of CK2α scores with the Ki-67 index in tumors, there was a positive correlation of higher CK2α score with higher percentage of Ki-67 positivity (P < 0.0003; Table 2).”

In Table 2 this is claimed as a Pearson chi-square test?

Page 14 Line 222 and following
=========================
“The average time frame for event occurrence was 3.2 years for progression free survival (PFS) and 3.6 years for overall survival (OS). PFS rates and OS rates among the CK2α scores were very similar, and OS was highest for patients with tumors having a CK2α score of 3 (P = 0.004, Table 2).”

What does average time mean in this context? Do these times take into account censoring? Is this the median as calculated from the KM analysis, or just an average of all survival times?

Why is a Pearson test used here when survival times generally should be analysed through survival analysis (KM or Cox regression)? I do not think the Pearson’s chi-square test is the best choice here. I would remove this section from Table 2.

Page 15 Line 231
==============

Please state exactly the multi-variate model fitted and associated HRs – could have this as a table. Did you check he proportional hazards assumption for the Cox models?

=====================================================

More detailed comments:
====================
The most interesting claimed result in the paper is:
“High CK2α protein levels are detected in HPV(+) OPSCC tumors and demonstrate an unexpected association with improved survival in a strongly HPV(+) OPSCC cohort.”

but we also know that

“the current increase in OPSCC incidence is considerably linked to HPV presence”

So HPV is related to an increase in OPSCC cancer but the associated increased CK2α improves overall survival, while for HPV(-) OPSCC an increased CK2α is associated with poorer overall survival.

Confounding is an issue all observational studies have to address, especially so when the results are unexpected. For any observational study the question is: Could bias or confounding explain the results?

This needs to be properly addressed in the paper to strengthen the claim. For example:

1. All the participants are male. So the results actually only apply to OPSCC cancer in men and not more generally. But more than this, gender is considered typically an important variable in these types of studies and the fact that it could not be included has weakened the results.

2. All participants come from one hospital, a potential section bias issue.

3. For the cohort none of the treatments appear to have an effect on overall survival (Table 3) – is this similar to reported clinical trial results for OPSCC? Are the HRs similar to the trials? If not then it is likely that the cohort is biased.

4. Were HPV(+) patients healthier than HPV(-) patients? Are there lifstyle differences? Could this or similar be the reason for the increase in overall survival. Was this investigated? Are there systematic differences between HPV(+) and HPV(-) patients in the cohort?

5. Treatment choice for the patients was likely not random – could this have an effect?

A convincing or at least plausible biological explanation is needed as an alternative. Have similar results been reported in the past?

Additional comments

I enjoyed reading the article and found it to be well written and an interesting study. My comments detailed above are mainly in the area of the role of possible bias and confounding in the study. The paper does not make any causal claims but I think that a robust analysis of the possibility of confounding needs to be included.

Reviewer 3 ·

Basic reporting

no comment

Experimental design

no comment

Validity of the findings

Given that 94% of patients were current or former smokers, is it really possible to "control" for smoking history in this patient population statistically? This shortcoming should be noted and the authors should suggest testing the external validity of their finding on a patient population with a more heterogeneous patient population in terms of smoking history.

Additional comments

The authors should explain their use of Ki-67 in the Introduction and provide their justification for inclusion of this marker in this study.

Lines 105-109 at the end of the Introduction read as results/discussion points.

How many patients were excluded due to simultaneous malignancy?

Were patients that had distant mets at presentation not excluded? Were patients that did not receive curative intent treatment (i.e. palliative treatment) also included in the analysis?

Line 160: consider adding "including those patients with discordant p16 results."

Line 185: add "or death." to the end of the sentence

Line 252: I believe the second HPV (+) was meant to be "HPV (-)"

Line 302: I'm not sure how effectively you can control for smoking statistically when 94% of patients have a smoking history.

Overall, this is a very organized and presented work with meaningful value to the head and neck oncology literature. I feel that with the above revisions, this article will make an excellent additional contribution to the current body of literature. I thank the authors for their submission.

---

## Round 0.2 · Minor Revisions

Could you kindly attend to the issue raised about Table I? As its not in the tracked changes version I am not sure if this has been amended.

Once that is dealt with I will recommend acceptance.

Thanks for the otherwise detailed, balanced and clear response to the points raised at review.

Reviewer 2 ·

Basic reporting

Changes or explanations as per the first review were made by the authors.

Experimental design

Changes or explanations as per the first review were made by the authors.

Validity of the findings

Limitations (possible bias and confounding) were discussed by the authors. I think this places the paper correctly in the literature.

Additional comments

The authors have addressed the issues I raised in as far as this is possible and have adequately discussed the issues with regard to possible bias and confounding.

One thing to note: In the pdf of the revised manuscript Table 1 does not look like it has been updated from the previous version.(The table is not included in the tracked word version).

---

## Round 0.3 · accepted · Accept

Thanks for this final change. Congratulations on a nice study.

[